# How Compatible Are Immune Checkpoint Inhibitors and Thermal Ablation for Liver Metastases?

**DOI:** 10.3390/cancers14092206

**Published:** 2022-04-28

**Authors:** Yasunori Minami, Haruyuki Takaki, Koichiro Yamakado, Masatoshi Kudo

**Affiliations:** 1Department of Gastroenterology and Hepatology, Faculty of Medicine, Kindai University, 377-2 Ohno-Higashi Osaka-Sayama, Osaka 589-8511, Japan; m-kudo@med.kindai.ac.jp; 2Department of Radiology, Hyogo College of Medicine, 1-1 Mukogawa Nishinomiya, Nishinomiya 663-8501, Japan; takakiharuyuki@gmail.com (H.T.); yamakado47@gmail.com (K.Y.)

**Keywords:** immune checkpoint inhibitor, immuno-oncology, information dissemination, liver metastasis, thermal ablation, tumor antigen

## Abstract

**Simple Summary:**

Although immune checkpoint inhibitors (ICIs) have achieved great progression in cancer treatment, the efficacy of ICI monotherapy is still limited. Meanwhile, the negative efficacy of thermal ablation for liver metastases is the high rate of local tumor progression. Since thermal ablation-induced inflammation and increases in tumor antigens have been suggested to promote the cancer-immunity cycle, thermal ablation and ICI can boost the immune response against cancer cells as one of the positive synergy effects. The findings of preclinical and clinical research have provided supportive evidence for the combination of ICIs with thermal ablation reversing T-cell exhaustion and demonstrating synergy. However, the clinical feasibility of immune response activation by combination therapy with ICI monotherapy and thermal ablation appears to be limited, it may be not very common phenomena.

**Abstract:**

Cancer immunotherapy, which reactivates the weakened immune cells of cancer patients, has achieved great success, and several immune checkpoint inhibitors (ICIs) are now available in clinical practice. Despite promising clinical outcomes, favorable responses are only observed in a fraction of patients, and resistance mechanisms, including the absence of tumor antigens, have been reported. Thermal ablation involves the induction of irreversible damage to cancer cells by localized heat and may result in the release of tumor antigens. The combination of immunotherapy and thermal ablation is an emerging therapeutic option with enhanced efficacy. Since thermal ablation-induced inflammation and increases in tumor antigens have been suggested to promote the cancer-immunity cycle, the combination of immuno-oncology (IO) therapy and thermal ablation may be mutually beneficial. In preclinical and clinical studies, the combination of ICI and thermal ablation significantly inhibited tumor growth, and synergistic antitumor effects appeared to prolong the survival of patients with secondary liver cancer. However, evidence for the efficacy of ICI monotherapy combined with thermal ablation is currently insufficient. Therefore, the clinical feasibility of immune response activation by ICI monotherapy combined with thermal ablation may be limited, and thermal ablation may be more compatible with dual ICIs (the IO–IO combination) to induce strong immune responses.

## 1. Introduction

Cancer immunotherapy has opened a new era in the field of oncology and may revolutionize cancer treatment for patients with advanced or metastatic solid tumors. Recent studies on immune checkpoint inhibitors (ICIs) demonstrated that their inhibitory effects prevented the blockade of the immune system by cancer cells attempting to escape their own destruction [1,2,3]. However, the efficacy of ICIs remains limited, and only a fraction of patients (<40%) respond to immuno-oncology (IO) drugs [4], which has been attributed not only to the complexity of the immune system, but also to it being a tightly regulated network [5]. Several resistance mechanisms have been identified and include the absence of tumor antigens, T-cell exhaustion in the tumor microenvironment (TME), and overexpression of β-catenin [6]. Therefore, the enhancement of antitumor immune responses, as a strategy to overcome resistance to ICIs, is urgently needed.

Although hepatic resection remains the only potentially curative option for patients with secondary liver cancer, minimally invasive locoregional therapies, including radiofrequency and microwave ablation, have been used to treat unresectable liver metastases. The majority of thermal ablation systems consist of a generator and needle electrode that delivers energy directly to the targeted tumor for cell death. Thermal ablation may induce local inflammation and release multiple biomolecules, such as tumor antigens, into the circulation. Locoregional thermal ablation has the potential to promote anticancer immune responses [7,8,9], and a very rare event associated with this therapy is the abscopal effect, which induces spontaneous distant tumor regression [10]. However, immunomodulatory synergy by combination therapies may induce systemically effective antitumor immunity. The findings of preclinical and clinical studies revealed that the combination of ICIs and thermal ablation significantly inhibited tumor growth, and synergistic antitumor effects appeared to prolong the survival of patients with secondary liver cancer. We herein discuss the efficacy of immunotherapy combined with thermal ablation.

## 2. Background

### 2.1. Immune Resistance and Tumor Antigens

Cancer cells are either resistant to immunotherapy (primary resistance) or initially respond and then develop resistance (acquired resistance) [11]. Resistance may also be classified as intrinsic or extrinsic to tumor cells [12]. Tumor-intrinsic factors include the depletion of neoantigens, defective antigen presentation, aberrant interferon signaling, tumor-induced exclusion/immunosuppression, and plasticity of tumor cells. Tumor-extrinsic factors include the lack of tumor T-cell infiltration and immunosuppression in the TME. The combination of tumor-intrinsic and -extrinsic factors induces a resistance to therapy, which leads to three different outcomes: (i) an insufficient antitumor T-cell response, (ii) the inactivation of tumor-specific T cells, and (iii) a weak memory T-cell response [13].

There are various mechanisms of immune resistance. At the first step of the cancer immunity cycle, the lack of sufficient tumor antigens and alterations in antigen processing and/or tumor antigen presentation have been implicated in impaired antitumor T-cell responses [14,15], and inadequate T-cell functions may induce various immunosuppressive reactions in the TME [16]. As one of the positive synergy effects, tumor antigen release by thermal ablation and monoclonal antibodies, targeting either the programmed cell death protein (PD-1) or its ligand PD-L1, can boost the immune response against cancer cells (Figure 1).

### 2.2. Impact of Thermally Induced Cell Death on Immune Responses

Thermal ablation is a localized treatment that induces tumor destruction by heating the tumor tissue to a core temperature > 100 °C [17,18]. This increase in temperature denatures intracellular proteins and causes the dissolution and melting of lipid bilayers, which irreversibly damage cancer cell membranes [19]. Temperatures > 105 °C strongly induce the transport of extracellular and intracellular water out of tissue, as well as tissue boiling, vaporization, and carbonization, which ultimately leads to cell death through coagulative necrosis [20]. During ablation, air bubbles produced by thermal ablation actively flow along the marginal hepatic vein to the main circulation.

A previous study examined sequential changes in serum carcinoembryonic antigen (CEA) levels in colorectal cancer patients treated with ablation for liver metastasis and showed that serum CEA levels initially increased in 65% of patients (11/17) and then gradually decreased to a nadir after 3 months [21]. Significant differences were observed in IL-2, IL-12, IL-1β, IL-8, and tumor necrosis factor-α levels 24 hours after ablation [22]. Endogenous danger signals, such as high mobility group box-1 and heat shock proteins, were also released from dying tumor cells. Furthermore, dying cells produced exosomes, which assisted in the maturation of dendritic cells and enhanced the tumor-specific cytotoxic T lymphocyte (CTL) response [23,24]. High levels of tumor-specific CD8+ T cells were previously shown to be induced in tumor-draining lymph nodes, due to heat-induced tumor damage [25,26]. Furthermore, antigen-presenting cells and tumor-infiltrating lymphocytes (TILs) were found to localize along the peripheral margin of the ablation zone [27]. Therefore, locoregional ablation therapy may provide a favorable TME for immune cells to function optimally, which will be beneficial when used in combination with ICIs [28,29].

## 3. Outcomes

### 3.1. Preclinical Studies

#### 3.1.1. Ablation-Induced Immunomodulation

Shi et al. retrospectively investigated thermal ablation-induced immune responses in the tumor tissues of colorectal cancer patients with liver metastasis [8]. They enrolled 38 patients who underwent preoperative thermal ablation for liver metastasis, followed by primary tumor resection (the ablation group), and 40 patients who underwent primary tumor resection alone (the non-ablation group). The number of TILs in the TME and degree of programmed death-ligand 1 (PD-L1) staining in tumor cells and lymphocytes were compared in colorectal cancer resection specimens. The median number of days between initial ablation for liver metastasis and surgical resection of the primary tumor was 6 (range, 4 to 10 days). Significantly higher levels of TILs were observed in the ablation group (*p* < 0.001), and the CD4/CD8 ratio was significantly higher in the ablation group (*p* = 0.002). These findings indicated that thermal ablation enhanced the tumor-specific immune responses and promoted the infiltration of CTLs into the TME.

#### 3.1.2. Combination of Ablation and Immunotherapy in Mice

Shi et al. also compared the outcomes of CT26 tumor-bearing mice treated with RFA alone, PD-1 antibody alone, and the combination of thermal ablation and the anti-PD-1 antibody [8]. The injection of ICIs was repeated on days 1, 4, 7, and 10 after ablation. Recurrence was not detected in the ablation zone in any group. Although the modest inhibition of tumor growth was observed in the mice treated with ablation alone or the anti-PD-1 antibody alone, the inhibition of tumor growth and prolonged survival were observed in the group treated with the combination of ablation and the anti-PD-1 antibody. Furthermore, CD8+ T-cell depletion eliminated the inhibition of tumor growth in the combination therapy group, indicating the further enhancement of CD8+ T cell-mediated antitumor immunity by the combination of ablation and PD-1.

Zhu et al. investigated the outcomes of the following groups of 4T1 breast cancer cell-bearing mice: no treatment, the combination of PD-1 and cytotoxic T lymphocyte-associated antigen (CTLA)-4 blockers, thermal ablation alone, and thermal ablation in combination with ICIs [30]. The mice were injected 3 and 6 days after ablation. The combination of PD-1 and CTLA-4 blockers did not significantly affect mouse survival. Survival was significantly longer in the thermal ablation in combination with ICIs group than in the no treatment, ICIs alone, or thermal ablation alone group (*p* < 0.001, *p* < 0.001, and *p* = 0.01, respectively). In parallel experiments, tumor-bearing mice were sacrificed on day 10 after ablation. Serum IFN-γ levels were significantly higher in the thermal ablation in combination with ICIs group than in the no treatment, ICIs alone, or thermal ablation alone groups (*p* < 0.001, *p* < 0.001, and *p* = 0.01, respectively). The 4T1 (*n* = 5) or CT26 colon adenocarcinoma (*n* = 5) cells were then implanted on the opposite flanks of ten surviving mice in the thermal ablation in combination with ICIs group. On day 25, after the rechallenge, the outgrowth of reimplanted 4T1 cells was significantly delayed (*p* = 0.002); however, no significant delays were observed for reimplanted CT26 cells (*p* = 0.905). Collectively, these findings indicated that the induction of tumor-specific immune responses are not elicited from prior exposure to different cancers.

### 3.2. Clinical Studies

#### 3.2.1. Oncological Outcomes

Since the initial approval of ipilimumab for the treatment of melanoma in 2011, several ICIs have been launched for cancer therapy. However, retrospective research on the combination of thermal ablation and ICIs has been limited.

McArthur et al. examined the tolerability of pre-operative cryoablation and/or ipilimumab in patients with operable breast cancer [31]. Patients were sequentially assigned into the following treatment groups: tumor cryoablation alone (*n* = 7), intravenous ipilimumab (10 mg/kg) alone (*n* = 6), and the combination of cryoablation and ipilimumab (*n* = 6). The cryoablation group was treated a median of 7 days before mastectomy (range, 4–10 days), and the ipilimumab group was treated a median of 10 days before mastectomy (range 8–13 days). The combination therapy group received ipilimumab a median of 3 days before undergoing cryoablation (range, 1–5 days). A peripheral blood flow cytometric analysis of the samples collected from the combination group revealed a sustained increase in the proportion of activated T cells, in both the CD4+ and CD8+ subsets. Furthermore, the proportion of Ki67-positive CD4+ T cells and CD8+ T cells was higher in the combination therapy group than in either therapy group alone. However, neither the proportion of CD4+ T cells, CD8+ T cells, or regulatory T cells, nor the ratio of effector T cells to regulatory T cells, significantly differed among the treatment groups. The ratio of Ki67-positive effector T cells to regulatory T cells within the tumors was also higher in the combination group (*p* = 0.05). However, the expression levels of immune checkpoint markers, including CTLA-4, PD-1, and lymphocyte activation gene-3, on CD4 and CD8 TILs, at the time of tumor resection, were similar among the treatment groups. Collectively, these findings revealed sustained peripheral increases in Th1-type cytokine levels, activated and proliferating CD4+ and CD8+ T cell numbers, and post-treatment proliferative T-effector cell numbers, relative to the T-regulatory cell numbers within tumors by combination therapy.

Prospective clinical trials have investigated the efficacy of thermal ablation combined with immunomodulation for various types of malignant tumors. A non-randomized phase II study, evaluating the abscopal effect of pembrolizumab after radiotherapy or thermal ablation in patients with mismatch repair-proficient metastatic colorectal cancer, is ongoing (NCT02437071) [32].

#### 3.2.2. Immune-Related Adverse Events (irAEs)

Leppelmann et al. retrospectively investigated whether interventional oncology procedures, performed in combination with anti–PD-1/PD-L1 therapy, increased the risk of irAEs in patients with different neoplasms [33]. Sixty-five patients (51% males; 63.1 ± 11.1 years) with melanoma (*n* = 28), non-small cell lung cancer (*n* = 12), hepatocellular carcinoma (*n* = 3), colorectal cancer (*n* = 2), and other malignancies were enrolled. They received anti–PD-1/PD-L1 agents ≤90 days before or ≤30 days after the interventional procedure, such as selective internal radiation therapy (*n* = 28), cryoablation (*n* = 22), thermal ablation (*n* = 16), and transarterial embolization (*n* = 12). The anti–PD-1/PD-L1 agents, pembrolizumab (*n* = 30), nivolumab (*n* = 22), and atezolizumab (*n* = 6), as well as the combination of ipilimumab and nivolumab (*n* = 7), were administered to patients. irAEs occurred in seven patients (10.8%), and the most common grade 1–2 complications were skin disorders (*n* = 5). Grade 3 irAEs, such as skin disorders (*n* = 1) and liver dysfunction (*n* = 1), were conservatively managed. The median time to the onset of irAEs was 33 days (interquartile range, 19–38 days). No unmanageable irAEs developed within 90 days of interventional treatments combined with ICIs.

#### 3.2.3. Information Dissemination Considering Scientific Credibility

Patient decisions about appropriate health care for specific clinical should be based on the best available current, valid, and relevant evidence. The internet makes finding evidence for clinical practice easy. Much evidence about thermal ablation and cancer immunotherapy can be found; however, it is difficult to define that sufficient appropriate evidence has been accumulated. Then, the concept of the information dissemination model can be useful for evaluating the spread of medical information. Information dissemination analysis provides a more detailed understanding of complex social dynamics [34,35,36,37]. According to a simplified model analysis, an increase in the credibility of scientific information and its relevance to the daily lives of individuals will enhance the speed and final scale of information spreading (Figure 2A). Figure 2B shows PubMed-indexed medical publications, in terms of hepatitis C and interferon, as an example of information dissemination. Interferon was formerly the only antiviral drug for chronic hepatitis C infection. Although PubMed-indexed medical publications had increased, until recently, interferon-based treatment has been declining, since the introduction of interferon-free therapy with direct-acting antivirals in 2014. With the terms “lung cancer” and “immune checkpoint inhibitor”, PubMed-indexed medical publications have markedly increased, due to the development of promising IO treatments, whereas those with the terms “thermal ablation” and “immune checkpoint inhibitor” have not (Figure 2C).

## 4. Summary

The efficacy of ICI monotherapy is still limited as a critical bottleneck, and an urgent medical question that needs to be addressed is how to convert an immune cold into a hot tumor. Meanwhile, the negative efficacy of thermal ablation for liver metastases is the high rate of local tumor progression [38,39]. There are each unmet needs for treatment efficacy. Here, we searched the manuscripts regarding synergistic effects in combination with IO and thermal ablation fairly. Among them, some preclinical/clinical manuscripts were selected, based on originality, logic, significance, and high impact on clinical application. Since thermal ablation-induced increases in tumor antigens have been suggested to promote the cancer-immunity cycle, the combination of IO therapy and thermal ablation may be mutually beneficial. The findings of preclinical and clinical research have provided supportive evidence for the combination of ICIs with thermal ablation reversing T-cell exhaustion and demonstrating clinical synergy; however, it cannot be very common phenomena. The speed of evidence accumulation on the combination of ICI monotherapy and thermal ablation has been insufficient. The clinical feasibility of immune response activation by combination therapy with ICI monotherapy and thermal ablation appears to be limited. Currently, combined immunotherapy is the treatment trend, and dual ICIs (IO–IO combination) are thought to prevent effector T-cell exhaustion, boosting the antitumor immune response powerfully. When synergistic effects in combination with ICI monotherapy are unacceptable, thermal ablation may be more compatible with dual ICIs (IO–IO combination) to induce strong immune responses.

## Figures and Tables

**Figure 1 cancers-14-02206-f001:**
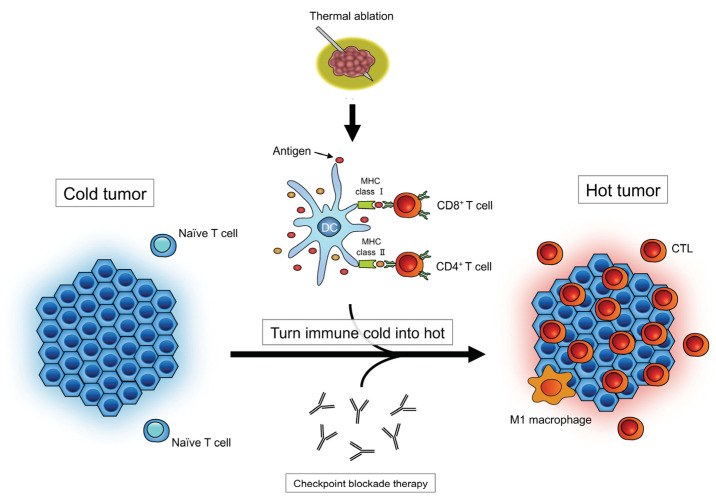
Activation of antitumor immune responses with thermal ablation. Note, CTL: cytotoxic T lymphocyte; DC: dendritic cell; MHC: major histocompatibility complex.

**Figure 2 cancers-14-02206-f002:**
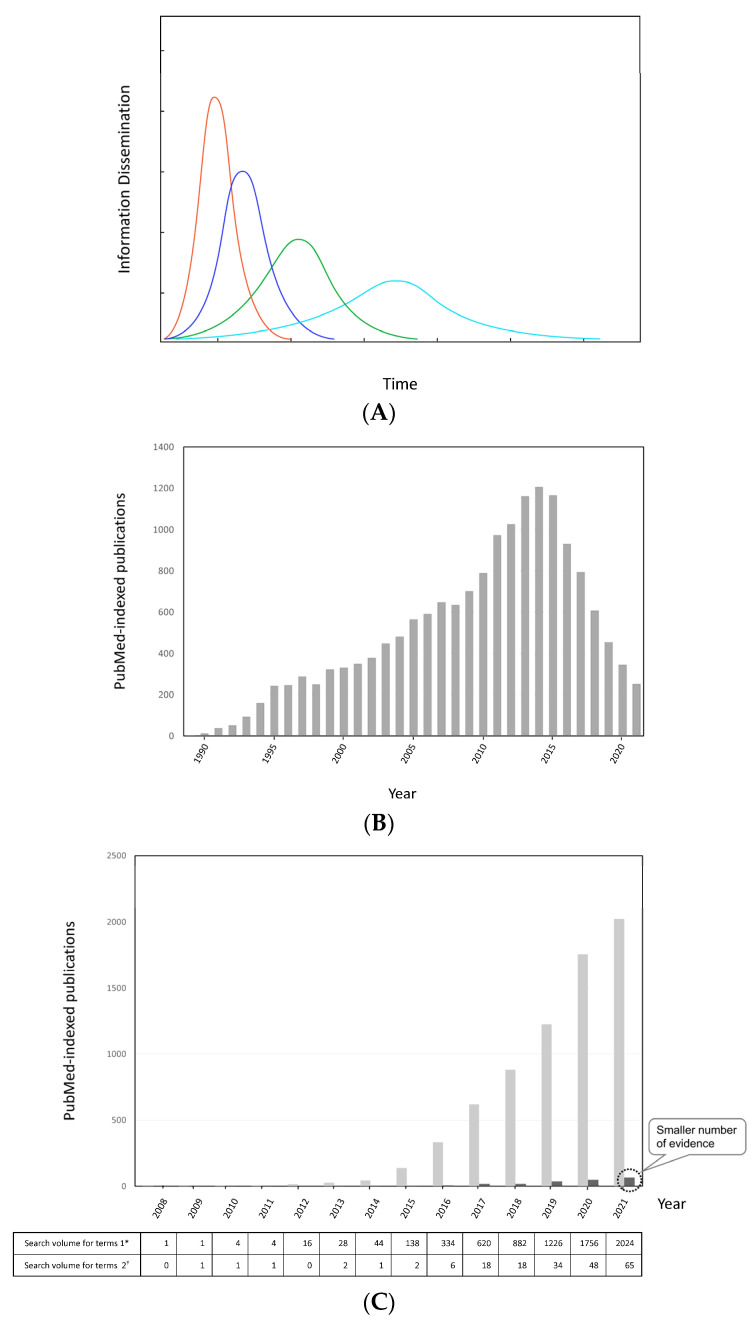
Information dissemination. (**A**) Simple model of rumor spreading. The stronger the correlation degree between medical information and real-world evidence is, the larger the spreading scale of medical information and faster the speed of dissemination are. Red line: The curve of information disemination when the credibility of scientific information was highest. Blue line: The curve of information disemination when the credibility of scientific information was 2nd higher. Green line: The curve of information disemination when the credibility of scientific information was tertiary. Light blue line: The curve of information disemination when the credibility of scientific information was lowest. (**B**) Annual research reports indexed in PubMed for the terms “hepatitis C” and “interferon”. (**C**) Annual research reports indexed in PubMed: “lung cancer and immune checkpoint inhibitor” vs. “thermal ablation and immune checkpoint inhibitor”. Note, search terms 1*: lung cancer and immune checkpoint inhibitor; search terms 2^†^: thermal ablation and immune checkpoint inhibitor. Light grey: The number of publications per year with topics of lung cancer and immune checkpoint inhibitor. Dark grey: The number of publications per year with topics of thermal ablation and immune checkpoint inhibitor.

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
