# Peer review of "How Compatible Are Immune Checkpoint Inhibitors and Thermal Ablation for Liver Metastases?"

_cancers, 2022, doi:10.3390/cancers14092206_

Round 1

Reviewer 1 Report

This is a timely topic relevant for cancer medicine. The cited publications are an incomplete selection of papers on this particular topic. Therefore, the authors should clarify why these (and not other papers on the same topic) papers were chosen. In order to strengthen the author`s point, a course of action should be suggested in the “Summary” of the manuscript.

The section on information dissemination is not really integrated in the argument. Its message is neither included in the abstract nor in the rationale (background) of the manuscript. The authors either remove this section or chose a different example to make their point.

Figure 1

The idea of the figures message is relevant. However, there are many more factors (not only CD4/8 T cells) contributing to the difference between “cold” and “hot” tumors. The authors should include more potential mechanisms into this figure.

Consider:

Insights into Imaging 2019; 10: 53

Semin Intervent Radiol 2019; 36: 334

Journal for ImmunoTherapy of Cancer 2017; 5:78

Author Response

Response to Reviewer #1

We wish to express our appreciation to the reviewer for providing insightful comments on our paper. The comments have helped us significantly improve the paper.

  1. This is a timely topic relevant for cancer medicine. The cited publications are an incomplete selection of papers on this particular topic. Therefore, the authors should clarify why these (and not other papers on the same topic) papers were chosen. In order to strengthen the author`s point, a course of action should be suggested in the “Summary” of the manuscript.

We revised the section of Summary as below,

The efficacy of ICI monotherapy is still limited as a critical bottleneck, and an urgent medical question that needs to be addressed is how to convert an immune cold into a hot tumor. Meanwhile, the negative efficacy of thermal ablation for liver metastases is the high rate of local tumor progression [38,39]. There are each unmet needs for treatment efficacy. Here, we searched the manuscripts about synergistic effects in combination with IO and thermal ablation fairly. Among them, some preclinical/clinical manuscripts were selected based on originality, logic, significance, and high impact on clinical application. Since thermal ablation-induced increases in tumor antigens have been suggested to promote the cancer-immunity cycle, the combination of IO therapy and thermal ablation may be mutually beneficial. The findings of preclinical and clinical research have provided supportive evidence for the combination of ICIs with thermal ablation reversing T-cell exhaustion and demonstrating clinical synergy; however, it can be not very common phenomena. The speed of evidence accumulation on the combination of ICI monotherapy and thermal ablation has been insufficient. The clinical feasibility of immune response activation by combination therapy with ICI monotherapy and thermal ablation appears to be limited. Currently, combined immunotherapy is the treatment trend, and dual ICIs (IO-IO combination) is thought to prevent effector T-cell exhaustion, boosting the antitumor immune response powerfully. When synergistic effects in combination with ICI monotherapy are unacceptable, thermal ablation may be more compatible with dual ICIs (IO-IO combination) to induce strong immune responses.

  1. The section on information dissemination is not really integrated in the argument. Its message is neither included in the abstract nor in the rationale (background) of the manuscript. The authors either remove this section or chose a different example to make their point.

We made a short comment in Abstract and Summary about insufficient evidence of synergistic effects in combination with ICI monotherapy and thermal ablation. We would like to explain the small volume of its evidence visually with Figure 2.  

We revised the subsection of “information dissemination” as below,

   Patient decisions about appropriate health care for specific clinical should be based on the best available current, valid and relevant evidence. The internet makes finding evidence for clinical practice easy. Much evidence about thermal ablation and cancer immunotherapy can be found; however, it is difficult to define that sufficient appropriate evidence has been accumulated. Then, the concept of information dissemination model can be useful to evaluate the spread of medical information. Information dissemination analysis provides a more detailed understanding of complex social dynamics[34-37]. According to a simplified model analysis, ….

  1. Figure 1

The idea of the figures message is relevant. However, there are many more factors (not only CD4/8 T cells) contributing to the difference between “cold” and “hot” tumors. The authors should include more potential mechanisms into this figure.

Consider:

Insights into Imaging 2019; 10: 53

Semin Intervent Radiol 2019; 36: 334

Journal for Immuno Therapy of Cancer 2017; 5:78

We modified the figure 1, and revised the 2nd paragraph in the subsection of “Immune resistance and tumor antigens” as below,

       There are various mechanisms of immune resistance. At the first step of cancer immunity cycle, the lack of sufficient tumor antigens and alterations in antigen processing and/or tumor antigen presentation have been implicated in impaired antitumor T-cell responses [14,15], and inadequate T-cell functions may induce various immunosuppressive reactions in the TME [16]. As one of positive synergy effects, tumor antigen release by thermal ablation and monoclonal antibodies targeting either the programmed cell death protein (PD-1) or its ligand PD-L1 can boost the immune response against cancer cells (Fig. 1).

Thank you again for your careful review of our manuscript. We look forward to receiving your further response.

Reviewer 2 Report

This manuscript is very interesting and valuable to the readers. Therefore, I am sure that the manuscript is fully worth being published.

I just recommend Figure 2 can be improved for readers to understand.

Author Response

Response to Reviewer #2

We wish to express our appreciation to the reviewer for providing insightful comments on our paper. The comments have helped us significantly improve the paper.

  1. This manuscript is very interesting and valuable to the readers. Therefore, I am sure that the manuscript is fully worth being published.

I just recommend Figure 2 can be improved for readers to understand.

We modified Figure 2c with adding a speech bubble and short comment.

Thank you again for your careful review of our manuscript. We look forward to receiving your further response.
